# Magnitude, outcome, and associated factors of anti-tuberculosis drug-induced hepatitis among tuberculosis patients in a tertiary hospital in North Ethiopia: A cross-sectional study

**Liwam Kidane Gezahegn**[1]*, **Ermias Argaw**[1], **Belete Assefa**[2], **Azeb Geberesilassie**[2], **Mengistu Hagazi**[2]

1 School of Medicine, College of Health Sciences, Mekelle University, Mekelle, Ethiopia, 2 School of Public Health, College of Health Sciences, Mekelle University, Mekelle, Ethiopia

* liwamey147@gmail.com

## Abstract

### Background

Short-course chemotherapy comprising isoniazid, rifampicin, ethambutol, and pyrazinamide has proved to be highly effective in the treatment of tuberculosis (TB). However, the most common adverse effect of this regimen leading to interruption of therapy is hepatotoxicity. There is a paucity of evidence in Tigray region on anti-tuberculosis drug-induced hepatitis. Therefore, this study aims to determine the magnitude, outcomes, and associated factors of drug-induced hepatitis in Ayder specialized comprehensive hospital tuberculosis clinic.

### Methods

An institution-based cross-sectional study was done on 188 cases of patients who took anti-tuberculosis drugs from August 4, 2015 to June 30, 2018 in tuberculosis clinic, Ayder Comprehensive Specialized Hospital, Northern Ethiopia. Data on socio-demography, clinical characteristics and magnitude of the incidence and outcome of anti-tuberculosis drugs-induced hepatitis were collected using structured checklist from patients' records using census method. Then, we entered and analyzed the data using statistical packages for social sciences (SPSS) statistical software version 21. Descriptive statistics were done in tables, counts, proportions, median and range. Bivariate and multivariable regression analyses were done to identify factors that are associated with drug-induced hepatitis. Confidence interval was taken at 95% and p-value of less than 0.05 was used to denote significance.

### Results

We approached a total of 226 patients' records, and we collected data from188 records (83.2%response rate). Anti-tuberculosis drug-induced hepatitis was found in 26 (13.8%) of the patients, of which 3 (11.54%) have died. Using multivariable logistic regression analyses, preexisting liver disease (AOR: 42.01, 95% CI: 4.22–417.49), taking other hepatotoxic

**Data Availability Statement:** All relevant data are within the manuscript and its Supporting Information files.

**Funding:** The cost of this research was covered by Mekelle University. The funder had no role in study design, data collection and analysis, decision to publish, or preparation of the manuscript.

**Competing interests:** The authors have declared that no competing interests exist.

drugs (AOR: 23.66, 95% CI: 1.77–314.79), and having lower serum albumin (AOR: 10.55, 95% CI: 2.57–43.32) were found to be significantly associated with the development of anti-tuberculosis drug-induced hepatitis.

## Conclusion and recommendation

The incidence of anti-tuberculosis drug-induced hepatitis was high. Patients with low baseline serum albumin, taking other hepatotoxic drugs and having preexisting liver disease should be followed with serial liver enzymes after initiation of anti-tuberculosis medications. These patients should be followed with frequent measurement of liver enzymes to assess for the development of drug-induced hepatitis.

## Background

Tuberculosis continues to be a major health problem in both developing and developed countries because of its resurgence among immunosuppressed patients [1–3]. Short-course chemotherapy comprising isoniazid (H), rifampicin (R), ethambutol (E), and pyrazinamide (Z) has proved to be highly effective in the treatment of TB (tuberculosis). However, the most common adverse effect leading to interruption of therapy is hepatotoxicity [4]. Multiple studies done in different countries reported that up to a quarter (2.55%- 36.75%) of patients taking anti-tuberculosis medications develop drug-induced hepatitis during the course of anti TB [5–13]. Anti-TB drug–induced hepatotoxicity is associated with a mortality of 6%– 12% if these drugs are continued after the onset of symptoms [14], and sometimes it can be as high as 22.7% [15]. There are many mechanisms for pathogenesis of drug-induced liver injury, but mostly the exact mechanism remains unclear. These include direct toxicity of the compound or its metabolite, free radicals, immune-mediated injury, and hypersensitivity reactions [16].

The occurrence of drug-induced hepatotoxicity is unpredictable, but it is observed that certain patients are at a relatively higher risk than other populations. Older age, female sex, chronic alcoholism and chronic liver disease, hepatitis B virus carrier status, acetylator status and nutritional status have all been incriminated as possible predisposing factors [17–20]. But data from different countries do not show consistent results, and therefore, it has been difficult to identify patients with increased risk of drug-induced hepatitis.

In Ethiopia, there are few studies of anti-tuberculosis induced hepatitis. Furthermore, there is a paucity of evidence in Tigray region on anti-tuberculosis drug-induced hepatitis. Therefore, this study aims to determine magnitude, outcome, and associated factors of drug-induced hepatitis in Ayder specialized comprehensive hospital tuberculosis clinic.

## Methods and materials

### Study design and setting

The study was conducted from May 1, 2018 to July 15, 2018 at Ayder comprehensive specialized hospital, located in Mekelle, capital of Tigray region, northern part of Ethiopia. It is the only tertiary care hospital in the region which gives service to more than 9 million people in its catchment area including the whole Tigray, some parts of Afar, North-eastern parts of Amhara region and Eritrean refugees. Tuberculosis (TB) diagnosis and treatment is one of the services given in the hospital. The TB clinic which is part of the infectious diseases' referral clinic has been functioning since 2008. It is run by one nurse and one internist.

## Study design

An institution-based retrospective cross-sectional study design.

## Inclusion and exclusion criteria

All patients' medical records who took anti TB medications at TB clinic, Ayder comprehensive specialized hospital from August 4, 2015 to June 30, 2018 were included. Patients who are below 18 years old, patients with incomplete clinical data, patients with incomplete investigations and patients who developed elevated liver enzymes and bilirubin while on treatment and in whom the elevation is explained with abnormal hepato-biliary imaging finding were excluded from this study.

## Sample size and sampling procedure

The assumptions used to calculate the actual sample size are: 95% level of confidence with 0.05 α value which yields Z α/2 = 1.96 on the standard normal distribution curve, 5% margin of error and using the incidence of anti-TB drug-induced Hepatitis 11.5% from a study done in Jimma, Ethiopia [21], to estimate the sample size using a single population proportion formula: n = $1.96^2$ x Pex (1- Pex)/$d^2$, where n = number of sample size: d = absolute precision (5%): Pex = Expected prevalence of anti-tuberculosis induced hepatitis taken as 11.5%. Using this formula, we found a minimum sample size of 156. However, we censused a total of 226 patients who were treated during the study period, of which, 188 patients fulfilled the inclusion criteria.

## Variables

**Dependent variables.**   The dependent variables are the incidence of anti-tuberculosis drug-induced hepatitis and outcome of drug-induced hepatitis.

**Independent variables.**   The independent variables are socio-demographic characteristics (Age, Gender), type of tuberculosis (extrapulmonary, pulmonary or both), diagnosis modality, severity of the tuberculosis, underlying chronic liver disease, chronic kidney disease, HBV infection status, HCV infection status, HIV infection status, CD4 count, having stage 4(AIDS) defining illness other than tuberculosis, using other hepatotoxic drugs, serum albumin, and baseline liver enzymes.

## Operational definition

**TB regimen:** In Ethiopia, the first line anti-TB drug formulations for adults are given in a fixed dose regimen, with HRZE (75mg/150mg/400mg/275mg) for intensive phase and HR (75mg/150mg) for continuation phase. The dose for adults is based on patients' weight band. Patients weighting 20-30kg take 1½tablet, patients weighting 30–39 kg take 2 tablets, patients weighting 40–54 kg take 3 tablets, and patients weighting 55kg and above take 4 tablets of the fixed dose regimen [22].

**Drug induced hepatitis:** Is defined based on American thoracic society's definition, if ALT and AST are at least 3 times the upper limit of the normal when symptoms of hepatitis are present or at least 5 times the upper limit of the normal if no symptoms of hepatitis are present [17, 23].

**Sever tuberculosis:** central nervous system (CNS) TB, pericardial TB, adrenal TB and miliary TB.

**Mild drug induced hepatitis:** Liver enzyme elevation 3–5 times [24].

**Moderate drug induced hepatitis:** Liver enzyme elevation 5–10 times [24].

**Severe drug induced hepatitis:** Liver enzyme elevation > 10 times [24].
Normal laboratory values based on ACSH laboratory's reference range:

**Normal AST:** For male up to 37 U/L and female up to 31

**Normal ALT:** For male up to 42 U/L and for female 32

**Normal bilirubin:** Total bilirubin 0.2–1.2 mg/dl and direct fraction of 0.2 mg/dl.

**Lower serum albumin:** Clinically significant low albumin level, less than 3.0 g/dl [25].

## Data collection instrument and technique

Data were collected using structured checklist prepared in English and designed to meet the study objectives. The data were collected from patients' charts. The checklist contained socio-demographic characteristics, disease-related factors, patient factors, including co-morbidities and outcome questions [6, 12, 26, 27].

Data collection was carried out using two data collectors who were selected from physicians of internal medicine and were given important training which guided them in collecting the appropriate data. Content validity of the checklist was assessed using experts and meanings of all items were checked accordingly.

## Data quality assurance

The checklist was also provided in the original language of English, avoiding the need for translation, and maintaining consistency of the questions and responses. Pre-test was conducted in 10% of the sample size to check the consistency of the checklist.

## Data analysis

The collected data were checked for completeness before the analysis process began. The checked data were coded and entered into Microsoft excel spreadsheet. The coded data were transferred to statistical packages for social sciences (SPSS) version 21. Descriptive statistics were done in tables, counts, proportions, median and range. Association of the independent variables with that of occurrence of anti-tuberculosis drugs induced hepatitis was checked using Bivariate and multivariable logistic regression. The statistical significance was determined at P value less than 0.05 and 95% confidence interval (CI).

## Ethical consideration

Ethical clearance was obtained from Ethical Review Committee of Mekelle University, College of health science institutional review board (IRB). Prior to data collection, official letter was obtained from department of internal medicine in order to get permission from the medical director and research office. The name of the patients was not mentioned and the entire information was kept for patient confidentiality.

## Result

### Socio-demography

We approached a total of 226 patients' records, and we collected data from 188 records (83.2% response rate). About 106 (56.4%) were males, about 33.5% aged between 18–29 years (Table 1).

Around 41% of the patients had both pulmonary and extrapulmonary tuberculosis, 62.8% of the patients' diagnosis of tuberculosis were made using clinical evidence and radiologic

**Table 1. Socio-demographic characteristics of study participants in Ayder comprehensive specialized hospital TB clinic, from August 4, 2015 to June 30, 2018 (N = 188).**

| Variable | Frequency | Percent (%) |
|---|---|---|
| **Sex** | | |
| Male | 106 | 56.4 |
| Female | 82 | 43.6 |
| **Age** | | |
| 18–29 | 63 | 33.5 |
| 30–39 | 45 | 23.9 |
| 40–49 | 37 | 19.7 |
| 50–59 | 21 | 11.2 |
| ≥60 | 22 | 11.7 |

modality. Around 80% of the patients had mild to moderate form of tuberculosis; about one-third used other hepatotoxic drugs. One-third of the patients had HIV co-morbidity, of these, 70.9% had stage 4 defining illness other than the tuberculosis; about three-forth had CD4 count less than 200. Around 26.6% had low serum albumin, 6.4% had pre-existing liver disease (cirrhosis, and moderate to severe fatty liver disease). About 81.9% of the patients had normal baseline liver enzymes, and 2.7% had Hepatitis B virus infection (Table 2).

## Magnitude of drug induced hepatitis

In three years, out of 188 patients, the magnitude of anti-tuberculosis drug-induced hepatitis was 13.8% and median time of onset for development of drug induced hepatitis was 12 days, ranging from 5–52 days. More than two-third of the patients developed moderate to severe type of drug-induced hepatitis and the most common symptom was nausea and vomiting. Eighty eight percent of the patients improved following discontinuation of the anti-tuberculosis drugs while the median time for normalization of the liver enzymes was 14 days, ranging 7–32 days. Among these patients, five had recurrence of drug-induced hepatitis after reinitiating of anti-tuberculosis drugs (Table 3).

## Bivariate analysis

On bivariate analysis, both descriptive analyses using Chi square and inferential analyses using binary logistic analysis was done. Type of tuberculosis, Severity of tuberculosis diagnosis, taking other hepatotoxic drugs, having preexisting chronic liver disease, HIV infection, lower serum albumin level, and abnormal baseline liver enzymes were significantly associated with the development of drug-induced hepatitis (Table 4).

## Multivariable analysis

In multivariable logistic regression analyses, having preexisting chronic liver disease, taking other hepatotoxic drugs and lower serum albumin were found to be independent predictors of developing drug-induced hepatitis. Patients with preexisting liver disease had 42.01 times higher risk of developing DIH (AOR: 42.01, 95% CI: 4.22–417.49). Patients taking other hepatotoxic drugs had 23.66 times higher risk of developing DIH (AOR: 23.66, 95% CI: 1.77–314.79). Patients having lower serum albumin had 10.55 higher risk of developing DIH (AOR: 10.55, 95% CI: 2.57–43.32) (Table 5).

**Table 2. Disease characteristics and co-morbidity of study participants in Ayder comprehensive specialized hospital TB clinic, from August 4, 2015 to June 30, 2018 (N = 188).**

| Variable | Frequency | Percent (%) |
|---|---|---|
| **Type of TB** | | |
| Extrapulmonary | 70 | 37.2 |
| Pulmonary | 41 | 21.8 |
| Both | 77 | 41.0 |
| **Diagnosis Modality** | | |
| Bacteriological | 28 | 14.9 |
| Histo-pathology | 40 | 21.3 |
| Radiology and Clinical | 118 | 62.8 |
| Clinical | 2 | 1.1 |
| **Severe Tuberculosis** | | |
| No | 150 | 79.8 |
| Yes | 38 | 20.2 |
| **Hepatotoxic Drugs**[*] | | |
| No | 131 | 69.7 |
| Yes | 57 | 30.3 |
| **HIV Status** | | |
| Positive | 55 | 29.3 |
| Negative | 113 | 60.1 |
| Unknown Status | 20 | 10.6 |
| **Stage 4 Defining Illness (n = 55)** | | |
| No | 16 | 29.1 |
| Yes | 39 | 70.9 |
| **CD4 Count (n = 55)** | | |
| $<200$cells/mm$^3$ | 41 | 74.5 |
| $\geq200$ cells/mm$^3$ | 14 | 25.5 |
| **Serum Albumin** | | |
| Low | 50 | 26.6 |
| Normal | 122 | 64.9 |
| Not Done | 16 | 8.5 |
| **Abdominal Ultrasound** | | |
| Pre-existing liver disease | 12 | 6.4 |
| No features of pre-existing liver disease | 154 | 81.9 |
| Not done | 22 | 11.7 |
| **Baseline Liver Enzymes** | | |
| Normal | 154 | 81.9 |
| Abnormal | 34 | 18.1 |
| **Hepatitis B Virus Infection** | | |
| Negative | 159 | 84.6 |
| Positive | 5 | 2.7 |
| Not done | 24 | 12.8 |

[*]Hepatotoxic drugs: Cotrimoxazole, fluconazole, atrovastatin, valporate, phenytoin, and propylthiouracil.

## Discussion

The magnitude of drug induced hepatitis was found to be 13.8%. Having preexisting chronic liver disease, taking other hepatotoxic drugs and lower serum albumin were found to be independent predictors of developing drug-induced hepatitis.

**Table 3. Magnitude of drug-induced hepatitis among study participants in Ayder comprehensive specialized hospital TB clinic, from August 4, 2015 to June 30, 2018 (N = 188).**

| Variable | Frequency | Percent (%) |
|---|---|---|
| **Anti-tuberculosis Drug-induced Hepatitis** | | |
| Yes | 26 | 13.8 |
| No | 162 | 86.2 |
| **Time of Onset of Anti-tuberculosis Drug-induced Hepatitis in Days (n = 26)** | | |
| Median (range) | 12 (5–52) | |
| **Severity of Drug Induced Hepatitis (n = 26)** | | |
| Mild | 8 | 30.8 |
| Moderate | 9 | 34.6 |
| Severe | 9 | 34.6 |
| **Clinical Feature of Patients with Anti-tuberculosis Drug-induced Hepatitis (n = 26)** | | |
| Nausea and vomiting | 25 | 96.2 |
| Malaise | 21 | 80.8 |
| Jaundice | 17 | 65.4 |
| **Outcome of Patients with Anti-tuberculosis Drug-induced Hepatitis (n = 26)** | | |
| Improved after discontinuation | 23 | 88.5 |
| Died | 3 | 11.5 |
| **Time to normalization of liver enzyme after discontinuation of anti-tuberculosis drugs in days (n = 23)** | | |
| Median (range) | 14 (7–32) | |
| **Recurrence of Anti-tuberculosis Drug-induced hepatitis after reinitiating (n = 23)** | | |
| No | 18 | 78.26 |
| Yes | 5 | 21.7 |

In this study, the magnitude of drug induced hepatitis was found to be high. Similar results were found in studies done in Pakistan, Egypt and Jimma, Ethiopia which were in the range of 11.5–15% [21, 28, 29]. This could be due to similar socio-demographic and economic background of the patients. In contrast to our finding, studies done in Malaysia, Nepal and Dawro Zone (South Ethiopia) showed the magnitude of DIH to be slightly lower, in the range of 8–9.4% [30–32]. This could be in part because the studies in Dawro (Ethiopia) and in Nepal were prospective cohort studies, and in the Malaysia the sample size was higher. In contrary to the current study, studies done in China and India indicated that the incidence to be very low, 2.55 and 3.8 respectively [5, 6]. The reason for this difference could be because these studies had a large sample size which was 3900 and 4304, respectively, and both were prospective studies.

The median time of development of the drug-induced hepatitis was 12 days, ranging 5–52 days. Other studies done in Pakistan, Egypt and Dawro zone (Ethiopia) had similar results [29, 31, 32].

The most common clinical features were nausea and vomiting (96.2%) followed by malaise and jaundice, 80.8% and 65.4%, respectively. This finding was in line with other studies in Jimma, (Ethiopia), Dawro Zone (Ethiopia), and Egypt [9, 21, 32]. This could be due to similarity of clinical features between acute viral hepatitis and drug-induced hepatitis.

In our study, we found that 34.6% of the patients had severe drug-induced hepatitis, which is similar with a study done in Jimma, Ethiopia [21].

In our study, age and sex were not significantly associated with development of DIH. Similar studies done in Iran and India also showed no statistically significant association between DIH and sex or age of the patients [26, 33]. In contrast, another study done in India found that

**Table 4. Bi-variate analysis of determinants of anti-tuberculosis drug-induced hepatitis in Ayder comprehensive specialized hospital, TB clinic, from August 4, 2015 to June 30, 2018.**

| Variable | COR | CI | | P value |
|---|---|---|---|---|
| | | Lower bound | Upper bound | |
| **Sex** | | | | |
| Male | 1.12 | 0.49 | 2.58 | 0.78 |
| Female | 1 | | | |
| **Age** | | | | |
| <60 | 1 | | | |
| ≥60 | 1.45 | 0.45 | 4.69 | 0.53 |
| **Type of Tuberculosis** | | | | |
| Extrapulmonary | 1 | | | |
| Pulmonary | 1.83 | 0.50 | 6.76 | 0.143 |
| Both | 3.81 | 1.31 | 11.06 | 0.01 |
| **Diagnosis Modality** | | | | |
| Bacteriological | 1 | | | |
| Radiology and clinical | 0.49 | 0.15 | 1.56 | 0.23 |
| Clinical | 4.40 | 0.91 | 21.24 | 0.06 |
| Histo-pathology | 0.59 | 0.15 | 2.28 | 0.45 |
| **Severe Tuberculosis** | | | | |
| No | 1 | | | |
| Yes | 4.48 | 1.86 | 10.78 | 0.001 |
| **Preexisting Liver Disease** | | | | |
| No | 1 | | | |
| Yes | 9.95 | 2.86 | 34.51 | 0.0001 |
| **Other Hepatotoxic Drugs** | | | | |
| No | 1 | | | |
| Yes | 15.24 | 5.36 | 43.31 | 0.0001 |
| **HIV Status** | | | | |
| Positive | 7.99 | 3.10 | 20.57 | 0.0001 |
| Negative | 1 | | | |
| **Stage 4 Defining Illness** | | | | |
| No | 1 | | | |
| Yes | 1.40 | 0.40 | 4.80 | 0.59 |
| **CD4 Count** | | | | |
| <200 cells/mm$^3$ | 2.24 | 0.66 | 7.54 | 0.19 |
| ≥200 cells/mm$^3$ | 1 | | | |
| **Low Serum Albumin** | | | | |
| No | 1 | | | |
| Yes | 17.06 | 5.97 | 48.72 | 0.000 |
| **Hepatitis B Virus Infection** | | | | |
| No | 1 | | | |
| Yes | 1.34 | 0.14 | 12.49 | 0.79 |
| **Abnormal Baseline Liver Enzymes** | | | | |
| No | 1 | | | |
| Yes | 4.43 | 1.81 | 10.84 | 0.001 |

patients with age ≥60 years and female gender were significantly associated with drug-induced hepatitis [6]. This could be due to participants in the Indian study were older as compared to our study, in which most of our study participants were younger age groups.

**Table 5. Multivariable regression of determinants of anti-tuberculosis drug-induced hepatitis in Ayder comprehensive specialized hospital TB clinic, from August4, 2015 to June 30, 2018.**

| Variable | COR | CI | | P value |
| --- | --- | --- | --- | --- |
| | | Lower bound | Upper bound | |
| **Type of Tuberculosis** | | | | |
| Extrapulmonary | 1 | | | |
| Pulmonary | 1.11 | 0.23 | 5.29 | 0.89 |
| Both | 1.62 | 0.26 | 9.84 | 0.60 |
| **Severe Tuberculosis** | | | | |
| No | 1 | | | |
| Yes | 0.75 | 0.18 | 3.08 | 0.69 |
| **Preexisting Liver Disease** | | | | |
| No | 1 | | | |
| Yes | 42.01 | 4.22 | 417.49 | 0.001 |
| **Other Hepatotoxic Drugs** | | | | |
| No | 1 | | | |
| Yes | 23.66 | 1.77 | 314.79 | 0.01 |
| **HIV Status** | | | | |
| Positive | 0.58 | 0.05 | 6.29 | 0.65 |
| Negative | 1 | | | |
| **Low Serum Albumin** | | | | |
| No | 1 | | | |
| Yes | 10.55 | 2.57 | 43.32 | 0.001 |
| **Abnormal Baseline Liver Enzymes** | | | | |
| No | 1 | | | |
| Yes | 1.53 | 0.41 | 5.60 | 0.52 |

In our study, type of tuberculosis, severity of the tuberculosis, and diagnostic modality were not significantly associated with drug-induced hepatitis. In contrast, in two studies done in India and one study in Jimma, Ethiopia showed that having extensive disease was significantly associated with the development of drug induced hepatitis [6, 21, 34]. In contrast to our study, in a study in India unconfirmed tuberculosis was significantly associated with development of DIH [35]. The reason for this could be the low number of patients who have definite diagnosis of tuberculosis in our study participants.

In our study, preexisting liver disease and use of other hepatotoxic drugs were significantly associated with DIH. Inline to the current study, use of other hepatotoxic drugs and preexisting liver disease were significantly associated with study done in Iran and India respectively [33, 35].

In this study, abnormal baseline liver enzyme was not significantly associated with DIH. Unlike our study, abnormal baseline liver enzyme was significantly associated with DIH in study done Iran [33].

This study showed that being HIV positive, lower CD4 count and stage 4 defining illness were not significantly associated with development of anti-tuberculosis drug-induced hepatitis. In contrary to our study, studies in Iran and Malaysia found HIV to be risk factor for DIH [33, 36]. The low number of HIV infected patients in the Malaysian study as compared to our study may explain this difference; in addition, the difference may be due to difference in study design as the Malaysian study was case-control.

In our study, having a positive HBsAg result was not significantly associated with development of drug-induced hepatitis. Similar to our study, in a study done in Malaysia hepatitis

virus infection was not associated with development of DIH. This may be due to the small number of patients with Hepatitis virus infection in both studies [30]. In contrast to our study, in a study done in India, hepatitis B infection was a significant risk factor for development of DIH [35]. This difference could be explained by the very low number of patients with HBsAg positive results in our study participants.

In our study participants, lower serum albumin was significantly associated with development of drug-induced hepatitis. This is in-line with studies done in India, Egypt and Malaysia [6, 29, 34, 36].

In our study, we found that among the patients who developed anti-tuberculosis drug-induced hepatitis, 23(88.5%) patients improved with discontinuation of the anti-tuberculosis treatment, and 3 (11.5%) patients died. Similarly, in a study done in China it was found that 3.77% died as result of DIH and 96.23% improved with drug discontinuation [5]. Similar results were also found in a study done in Egypt [29].

In our study, no patient developed acute liver failure. In contrast to our study, a study done in India found that one-fourth of the patients developed serious complications, such as fulminant and subacute hepatic failure [35]. The difference might be because our study population was on direct observed therapy; therefore, early detecting of the DIH and discontinuation of the drugs might contribute to this low incidence of acute liver failure.

The median time to normalization of liver enzymes was 14 days, ranging from 7-30 days. Similar results were found in a study done in Egypt [29]. In our study, recurrence of drug-induced was observed in 5 (21.7%) of the patients, which were similar with studies done in India and New York [6, 27].

## Limitations of the study

Important variables, like alcohol intake and body mass index were not included in the study due to the retrospective nature of the study. Another limitation was the small sample used; therefore, generalizing the result is difficult.

## Conclusion

In this study the magnitude of drug induced hepatitis was found to be high. Lower serum albumin, taking other hepatotoxic drugs, and having preexisting chronic liver disease were found to be independent predictors of drug-induced hepatitis. There was high rate of recurrence and mortality.

## Supporting information

**S1 Dataset.**
(SAV)

**S1 File.**
(SAV)

## Acknowledgments

We would like to thank all the card room staff and tuberculosis clinic nurse for helping us. In addition, we are thankful to our friends who gave us their precious time helping throughout this work.

## Author Contributions

**Conceptualization:** Liwam Kidane Gezahegn, Ermias Argaw, Azeb Geberesilassie, Mengistu Hagazi.

**Data curation:** Liwam Kidane Gezahegn, Ermias Argaw.

**Formal analysis:** Liwam Kidane Gezahegn, Ermias Argaw, Belete Assefa.

**Funding acquisition:** Liwam Kidane Gezahegn, Ermias Argaw.

**Investigation:** Liwam Kidane Gezahegn, Ermias Argaw, Belete Assefa, Azeb Geberesilassie.

**Methodology:** Liwam Kidane Gezahegn, Ermias Argaw, Belete Assefa, Azeb Geberesilassie.

**Project administration:** Liwam Kidane Gezahegn.

**Resources:** Liwam Kidane Gezahegn.

**Software:** Liwam Kidane Gezahegn, Belete Assefa.

**Supervision:** Liwam Kidane Gezahegn, Ermias Argaw, Belete Assefa.

**Validation:** Liwam Kidane Gezahegn, Ermias Argaw, Belete Assefa, Azeb Geberesilassie.

**Visualization:** Liwam Kidane Gezahegn, Ermias Argaw, Belete Assefa, Azeb Geberesilassie.

**Writing – original draft:** Liwam Kidane Gezahegn, Mengistu Hagazi.

**Writing – review & editing:** Liwam Kidane Gezahegn, Ermias Argaw, Belete Assefa, Azeb Geberesilassie, Mengistu Hagazi.

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
