## [Decision Letter · Decision Letter 0]

17 Sep 2020

PONE-D-20-23615

Magnitude, Outcome, and Associated Factors of Anti-tuberculosis Drug-induced Hepatitis among Tuberculosis patients in a Tertiary hospital in North Ethiopia: A Cross-sectional study

PLOS ONE

Dear Dr. gezahegn,

Thank you for submitting your manuscript to PLOS ONE. After careful consideration, we feel that it has merit but does not fully meet PLOS ONE’s publication criteria as it currently stands. Therefore, we invite you to submit a revised version of the manuscript that addresses the points raised during the review process.

We look forward to receiving your revised manuscript.

Kind regards,

Bin Su, Ph.D.

Academic Editor

PLOS ONE

Journal Requirements:

2. In ethics statement in the manuscript and in the online submission form, please provide additional information about the patient records used in your retrospective study. Specifically, please ensure that you have discussed whether all data were fully anonymized before you accessed them and/or whether the IRB or ethics committee waived the requirement for informed consent. If patients provided informed written consent to have data from their medical records used in research, please include this information.

4. We note you have included a table to which you do not refer in the text of your manuscript. Please ensure that you refer to Table 3 in your text; if accepted, production will need this reference to link the reader to the Table.

Reviewers' comments:

Reviewer's Responses to Questions

**Comments to the Author**

1. Is the manuscript technically sound, and do the data support the conclusions?

Reviewer #1: No

Reviewer #2: Yes

2. Has the statistical analysis been performed appropriately and rigorously? 

Reviewer #1: I Don't Know

Reviewer #2: Yes

3. Have the authors made all data underlying the findings in their manuscript fully available?

Reviewer #1: No

Reviewer #2: Yes

4. Is the manuscript presented in an intelligible fashion and written in standard English?

Reviewer #1: No

Reviewer #2: Yes

5. Review Comments to the Author

Reviewer #1: The authors described a an institution-based cross-sectional study of DILI due to short course chemotherapy in the treatment of tuberculosis (TB). There are several other reported studies with more sophisticated designs for the same purpose. The overall quality of the presented clinical study is limited by the nature of the study design. The authors may also want to improve the preparation of the manuscript.

1. Small sample size as a cross-sectional study limited the power of the study.

2. More clinical details need to provided. For example,

a. Details of drug combination and dosing regimen need to be added. This is important because drug-drug interaction is a known factor for DILI of the chemotherapy for TB. The details of drug combination and dosing regimen are critical to understand the risk of DILI due to DDI.

b. What are "other hepatotoxic drugs" discussed in the manuscript?

c. What are the "preexisting liver disease" discussed in the manuscript? The severity of the preexisting conditions?

3. Language quality of the manuscript need to be improved.

a. Many spaces are missing cause readability issues.

b. Scientific and accurate description is preferred.

Reviewer #2: The manuscript by Liwam et al. describes the importance, outcome, and the other factors associated with anti-tuberculosis drug-induced hepatitis among tuberculosis patients. Tuberculosis is one of the major global health issues, which is associated with hepatitis as an adverse effect causing interruption to the treatment. Since the drug induces hepatotoxicity is unpredictable, it is necessary to identify and categorize patients into different risk levels. The study has shown that patients with low baseline serum albumin, taking other hepatotoxic drugs, and having the preexisting liver disease are at high risk of anti-TB drug induces hepatitis. The authors made an interesting study and have collected and considered a unique dataset from the cross-sectional study. The paper is generally well written and structured. However, in my opinion, attention should be given to the following issues described below.

1. Data from Table-2, for calculating frequency and percentage for Stage 4 defining illness and CD4 count, the total participants number N, will be 55 (HIV positive status) and not 188. The calculations were made using N=55; So, the authors are encouraged to mention the N value specifically.

2. Grammar and spacing errors are present in the manuscript; the authors should carefully proofread the paper for language errors.

6. PLOS authors have the option to publish the peer review history of their article (what does this mean?). If published, this will include your full peer review and any attached files.

Reviewer #1: No

Reviewer #2: No

---

## [Author Response · Author response to Decision Letter 0]

9 Oct 2020

To the academic editor 

Response: We followed PLOS ONE's style requirements,

2. In ethics statement in the manuscript and in the online submission form, please provide additional information about the patient records used in your retrospective study. Specifically, please ensure that you have discussed whether all data were fully anonymized before you accessed them and/or whether the IRB or ethics committee waived the requirement for informed consent. If patients provided informed written consent to have data from their medical records used in research, please include this information

Response: We have already described that the identification of patients was not mentioned and the entire information was kept for patient confidentiality.

3. PLOS requires an ORCID iD for the corresponding author in Editorial Manager on papers submitted after December 6th, 2016. Please ensure that you have an ORCID iD and that it is validated in Editorial Manager.

Response: 0000-0002-1378-0670

4. We note you have included a table to which you do not refer in the text of your manuscript. Please ensure that you refer to Table 3 in your text; 

Response: We have included “Table 3” in the text of our manuscript 

5. Please include captions for your Supporting Information files at the end of your manuscript, and update any in-text citations to match accordingly.

Response: We included “S1 dataset” as supporting information.

To reviewer 1

1. Small sample size as a cross-sectional study limited the power of the study.

Response: comment accepted.

2. More clinical details need to provided. For example,

a. Details of drug combination and dosing regimen need to be added. This is important because drug-drug interaction is a known factor for DILI of the chemotherapy for TB. The details of drug combination and dosing regimen are critical to understand the risk of DILI due to DDI.

Response: We included the following information:

“TB regimen; In Ethiopia, the first line anti-TB drug formulations for adults are given in a fixed dose regimen, with HRZE (75mg/150mg/400mg/275mg) for intensive phase and HR (75mg/150mg) for continuation phase. The dose for adults is based on patients’ weight band. Patients weighting 20-30kg take 1½ tablet, patients weighting 30-39 kg take 2 tablets, patients weighting 40-54 kg take 3 tablets and patients weighting 55kg and above take 4 tablets of the fixed dose regimen”

 b. What are "other hepatotoxic drugs" discussed in the manuscript?

Response: we included the following information under table 2 as a footnote: 

“*Hepato-toxic drugs: Cotrimoxazole, fluconazole, atrovastatin, valporate, phenytoin, and propylthiouracil”

c. What are the "preexisting liver disease" discussed in the manuscript? The severity of the preexisting conditions?

Response: we inserted the following information in the text description section of table 2: 

“(Cirrhosis, and moderate to severe fatty liver disease)”

3. Language quality of the manuscript need to be improved.

a. Many spaces are missing cause readability issues.

Response: We went through end to end of the manuscript and corrected the formatting and grammatical errors as it is evidenced in the document track changes. 

b. Scientific and accurate description is preferred.

Response: we tried to follow scientific writing style and we corrected the errors in the table. 

To reviewer 2

1. Data from Table-2, for calculating frequency and percentage for Stage 4 defining illness and CD4 count, the total participants number N, will be 55 (HIV positive status) and not 188. The calculations were made using N=55; So, the authors are encouraged to mention the N value specifically.

 Response: corrected accordingly 

2. Grammar and spacing errors are present in the manuscript; the authors should carefully proofread the paper for language errors.

Response: We went through end to end of the manuscript and corrected the formatting and grammatical errors as it is evidenced in the document track changes.

---

## [Editor Report · Decision Letter 1]

14 Oct 2020

Magnitude, Outcome, and Associated Factors of Anti-tuberculosis Drug-induced Hepatitis among Tuberculosis patients in a Tertiary hospital in North Ethiopia: A Cross-sectional study

PONE-D-20-23615R1

Dear Dr. %Gezahegn%,

We’re pleased to inform you that your manuscript has been judged scientifically suitable for publication and will be formally accepted for publication once it meets all outstanding technical requirements.

Kind regards,

Bin Su, Ph.D.

Academic Editor

PLOS ONE
---

## [Editor Report · Acceptance letter]

23 Oct 2020

PONE-D-20-23615R1 

Magnitude, Outcome, and Associated Factors of Anti-tuberculosis Drug-induced Hepatitis among Tuberculosis Patients in a Tertiary Hospital in North Ethiopia: A Cross-sectional Study 

Dear Dr. Gezahegn:

I'm pleased to inform you that your manuscript has been deemed suitable for publication in PLOS ONE. Congratulations! Your manuscript is now with our production department. 

Kind regards, 

on behalf of

Dr. Bin Su 

Academic Editor

PLOS ONE